# Prediction of Food Safety Risk Level of Wheat in China Based on Pyraformer Neural Network Model for Heavy Metal Contamination

**DOI:** 10.3390/foods12091843

**Published:** 2023-04-29

**Authors:** Wei Dong, Tianyu Hu, Qingchuan Zhang, Furong Deng, Mengyao Wang, Jianlei Kong, Yishu Dai

**Affiliations:** 1National Engineering Research Centre for Agri-Product Quality Traceability, Beijing Technology and Business University, Beijing 100048, China; dongwei2019@btbu.edu.cn (W.D.); hu2293425982@163.com (T.H.); zqc1982@126.com (Q.Z.); 2231101003@st.btbu.edu.cn (F.D.); wangmengyao618@163.com (M.W.); kongjianlei@btbu.edu.cn (J.K.); 2China Food Flavor and Nutrition Health Innovation Center, Beijing Technology and Business University, Beijing 100048, China; 3School of E-Business and Logistics, Beijing Technology and Business University, Beijing 100048, China; 4School of Artificial Intelligence, Beijing Technology and Business University, Beijing 100048, China

**Keywords:** wheat, food safety risk level, heavy metal contamination, Pyraformer neural network model

## Abstract

Heavy metal contamination in wheat not only endangers human health, but also causes crop quality degradation, leads to economic losses and affects social stability. Therefore, this paper proposes a Pyraformer-based model to predict the safety risk level of Chinese wheat contaminated with heavy metals. First, based on the heavy metal sampling data of wheat and the dietary consumption data of residents, a wheat risk level dataset was constructed using the risk evaluation method; a data-driven approach was used to classify the dataset into risk levels using the K-Means++ clustering algorithm; and, finally, on the constructed dataset, Pyraformer was used to predict the risk assessment indicator and, thus, the risk level. In this paper, the proposed model was compared to the constructed dataset, and for the dataset with the lowest risk level, the precision and recall of this model still reached more than 90%, which was 25.38–4.15% and 18.42–5.26% higher, respectively. The model proposed in this paper provides a technical means for hierarchical management and early warning of heavy metal contamination of wheat in China, and also provides a scientific basis for dynamic monitoring and integrated prevention of heavy metal contamination of wheat in farmland.

## 1. Introduction

As one of the three major food crops in the world today, wheat plays an important role in the diets of most countries around the world, and is also a major food crop in China. However, with the long-term unreasonable application of pesticides, fertilizers and mulch; industrial production; automobile exhaust emissions; and irrigation of industrial wastewater, the content of heavy metals in the soil and water environment has increased. The elimination of heavy metal pollution is often more difficult because heavy metals cannot be microbially or chemically degraded, and can only undergo interconversion between various forms in the environment [1]. After heavy metals pollute the atmosphere, water bodies and soil, they are enriched in the human body at the end of the food chain, endangering human health, leading to metabolic disorders in the organism and inducing diseases and even death [2]. Among a series of heavy metals, mercury (Hg), cadmium (Cd), arsenic (As), lead (Pb) and chromium (Cr) are known as the five toxic elements because they are the most toxic. Among them, Hg, Cd, As and Pb are harmful to organisms in many ways as non-essential substances for the metabolism and other biological functions [3], while Cr is one of the essential trace elements for humans and animals, but excessive intake can have significant toxic effects on humans. Studies have shown that Pb, when entering the human body, can cause abnormalities in the function of many systems, such as the hematopoietic system, the nervous system and the immune system. It can also cause liver, kidney, gastrointestinal and brain diseases, in aerosol or liquid form, diffuse in the daily air environment [4,5]. Cd is more easily adsorbed by crops compared to other heavy metal elements, and after enrichment in the human body, it impairs bone metabolism and leads to osteochondrosis. In addition, chronic Cd poisoning has an effect on human fertility [6]. Hg is one of the most toxic heavy metal elements in the environment; it directly sinks into the liver after being ingested, causing great damage to the brain, nerves and vision [7,8]. The toxicity of As is related to the form and valence state in which it is present, and the toxicity of arsenicals increases steeply. Chronic As poisoning can induce lung, skin and bladder cancer, while acute poisoning can lead to death within days or even hours [9,10]. Cr compounds are tremendously harmful to humans, and studies have found that trivalent Cr is teratogenic, while hexavalent Cr is far more toxic than trivalent Cr. It is also a strong carcinogen which can induce lung and nasopharyngeal cancer [11].

Besides being hazardous to human health, heavy metal contamination can also cause crop quality degradation, which can lead to serious economic losses and affect national economic development. Therefore, in view of the potential consequences of heavy metal contamination, a study on the prediction of the safety risk level of heavy metal contamination in wheat, which is the staple food of the nation, is imminent.

Mao et al. [12] compared and analyzed the monitoring of heavy metal contamination of food products at home and abroad, and found that foreign countries monitored the “farm to table” process of determining the quality of food earlier, while China had problems such as difficulties in the precise tracing of factors including the origin environment, planting and cultivation measures and incomplete coverage of heavy metal indexes. Mahmoud et al. [13] evaluated the ecological risk of Cd, As, Cr and Pb metals in wheat raised near industrial parks, and showed that no non-carcinogenic risk was found in the studied population through the four exposure routes; however, the carcinogenic risk of Cd, As and Pb was considerable through oral consumption of wheat and ingestion of soil. Li et al. [14] analyzed the characteristics of heavy metal content in wheat grains and assessed the human health risks in a county in northern Henan Province, showing that the minimum health risk indicator for As, Cd and Cr was 6.32 × 10^−4^ for adults and children, which exceeded the maximum acceptable risk range recommended by the U.S. Environmental Protection Agency (EPA) and posed a high carcinogenic risk. Doabi et al. [15] studied heavy metal contamination and health risks in agricultural soils, atmospheric dust and major food crops in Kermanshah province, Iran, where 167 samples of agricultural soils, atmospheric dust and food crops (wheat and maize) were collected and analyzed for concentrations, contamination levels and human health risks of four heavy metals, including zinc (Zn), copper (Cu), nickel (Ni) and Cr, respectively. Numerous studies have shown that the degree of heavy metal contamination varies widely from region to region, and that the types of heavy metal contamination in different regions vary. This shows that pollution prevention and timely warnings for agricultural products appear to be necessary for their targeted management and regulation. Voronenko et al. [16] analyzed and predicted the food security indicator for Ukraine and predicted the level of risk of its decline using data from the time interval of 1995 to 2018. Kim [17] et al. used the Integrated Fisheries Risk Analysis Method for Ecosystems (IFRAME) to assess and predict risk indices at dynamic and spatial-temporal scales in the Korean ecosystem in order to establish an appropriate fishery management plan for sustainable fisheries by incorporating spatial variability into the ecosystem. Tavoloni et al. [18] performed temporal trend analysis and predicted the heavy metal levels in clams and mussels as a way to assess environmental safety, which allowed for the assessment of contaminants in water bodies and sediments. Lu et al. [19] used deep learning algorithms to assess and predict the risk of contamination levels of Cd, Cr and As metals and metalloids in Chinese rice. The vast area of China, the difference in climate between the north and the south, the difference in economic development between the east and the west and the difference in soil topography and geology were expected to lead to heavy metal pollution in soil, showing the characteristics of geological factors [20]. In addition, studies have shown that global climate change is further influencing the diffusion of heavy metals in groundwater [21]. Meanwhile, studies by the U.S. Geological Survey and Harvard University have shown that a series of seemingly unrelated changes triggered by global climate change will have a significant impact on the transport and distribution of the heavy metal mercury on a global scale.

In summary, the graded supervision and risk warning of heavy metal pollution in wheat in each province of China can not only strengthen the supervision in key areas, but also provide a theoretical basis for the monitoring and comprehensive management of heavy metal pollution in farmland wheat. In this study, a Pyraformer-based model was constructed to predict the safety risk level of heavy metal contamination in wheat in China.

Using the sampling data of heavy metal content in national wheat samples by the State Administration for Market Regulation, combined with the Fifth China Total Diet Study [22], the food safety risk evaluation indicator dataset of wheat was constructed and the K-Means++ algorithm was used to divide the space constructed from the indicator dataset into the corresponding risk levels. On the constructed dataset, the Pyraformer neural network model was used to predict the assessment indicator based on the constructed time series, thus completing the prediction of the wheat risk level. The model proposed in this paper provides a technical means for implementing hierarchical management of each risk area, focusing on key areas, early warning and early treatment.

## 2. Materials and Methods

### 2.1. Sample Collection and Processing

The data in this study were obtained from the content of heavy metals in wheat samples taken by the State Administration for Market Regulation from 2019 to 2021, covering 20 provinces, with a total of 72,254 wheat samples collected. Among them, the contents of Pb, Cd and Cr were determined by graphite furnace atomic absorption spectrometry, As by inductively coupled plasma mass spectrometry and Hg by cold atomic absorption spectrometry.

### 2.2. Evaluation Methods and Criteria

According to the statistical analysis of the sampling data from the State Administration for Market Regulation, the proportion of wheat samples without detectable heavy metal contamination in this study was less than 60% of the total samples. Therefore, according to the principle of credible evaluation of low-level contaminants in food proposed by the WHO, the data not detected in this experiment were assigned a 1/2 LOD (limit of detection) value. The five heavy metal elements, Hg, Cd, As, Pb and Cr, in this study all have chronic non-carcinogenic health risks, with Cr, As, Cd and Pb having carcinogenic risks.

#### 2.2.1. Nemerow Integrated Pollution Index

The Nemerow Integrated Pollution Indicator (NIPI) is a weighted multi-factor environmental quality indicator that takes into account extreme values, and is one of the most common methods used at home and abroad to analyze the pollution levels in soil [23,24], crop [25,26], water [27,28,29], atmosphere [30,31], fruits [32] and vegetables [33]. It can take into account the most polluting impact factors and objectively reflect the comprehensive impact of various pollutants on wheat. Among them, the degree of contamination of a single heavy metal was assessed by the single-factor contamination indicator method, and the expression is shown below:(1)$$P_{i,j}=\frac{C_{i,j}}{S_{j}}$$

In this paper, we regard the *i*th as the i-th in the terminology, and other forms of xth in the following have the same meaning as this.

Where $P_{i,j}$ is the single factor contamination indicator of the *j*th heavy metal contaminant in wheat from province *i*, $C_{i,j}$ is the detected level of the *j*th heavy metal contaminant in wheat from province *i* (mg/kg) and $S_{j}$ is the limit value (mg/kg) specified in the national standard for the *j*th heavy metal contaminant in wheat. The National Standard for Food Safety Limits for Contaminants in Food provides limits on heavy metals in wheat in China, with the limits of total Hg, Cd, total As, Pb and Cr being 0.02, 0.1, 0.5, 0.2 and 1.0 mg/kg, respectively.
(2)$${\mathrm{PI}}_{i}=\sqrt{\frac{P_{max}^{2}(i)+P_{\mathrm{avg}}^{2}\left(i\right)}{2}}$$
where ${\mathrm{PI}}_{i}$ is the NIPI of heavy metal contamination in wheat of province i; $P_{\max}\left(i\right)$ is the maximum single-factor pollution indicator of heavy metal contaminants in wheat of province i and $P_{\mathrm{avg}}(i)$ is the average pollution indicator of heavy metal contaminants in wheat of province i. Usually, the single-factor contamination indicator and the NIPI are greater than 1, indicating that the product is contaminated, or less than or equal to 1, indicating that it is not contaminated, and the greater the value is, the more serious the contamination is.

#### 2.2.2. Target Hazard Quotient

In this study, the target hazard quotient (THQ) proposed by the U.S. EPA was used to assess the chronic non-carcinogenic health risk to the body from heavy metal contaminants ingested by humans through wheat consumption [34,35]. When more than one heavy metal contaminant is present in wheat, the combined effect of the contaminants should be considered. Therefore, in order to assess the risk of multiple heavy metal contaminants to human health, the total target hazard quotient was used in this paper to calculate the sum of the target hazard quotients of five heavy metal contaminants in wheat. When THQ is less than 1, it indicates that there is no significant health risk to the human body from heavy metal contaminants ingested through wheat, and when THQ is greater than or equal to 1, it indicates that the contamination poses a certain health risk to human body; the greater the THQ, the greater the health risk. The expression of the target hazard quotient is shown below:(3)$${\mathrm{THQ}}_{i,j}=\frac{{\mathrm{EF}\times \mathrm{ED}\times F}_{i}^{50}\times C_{\mathrm{avg}(i,j)}}{\mathrm{RfD}\times W\times \mathrm{AT}}$$
where ${\mathrm{THQ}}_{i,j}$ is the THQ of the jth heavy metal pollutant in wheat in province i; EF is the frequency of wheat intake per year (365 days/year); ED is the intake time of wheat, which takes the value of 70 years; $F_{i}^{50}$ is the average intake of wheat for residents of province i (kg/d); $C_{\mathrm{avg}(i,j)}$ is the average content of the jth heavy metal contaminant in wheat in province i (mg/kg); RfD is the reference dose (mg/(kg·d)) of exposure to heavy metal contaminants via the ingestion route; W is the average weight of the inhabitants (kg), taken as 60 kg; and AT is the average exposure time (365 days/year×ED). EPA provides the average daily reference dose of heavy metals, where the RfD of Hg, Cd, As, Pb and Cr are 0.0001, 0.001, 0.0003, 0.0037 and 0.003 mg/(kg·d), respectively.
(4)$${\mathrm{THQ}}_{i}=\sum_{j}{\mathrm{THQ}}_{i,j}$$
where ${\mathrm{THQ}}_{i}$ is the total target hazard quotient of heavy metal pollutants in wheat in province i, which is the sum of the target hazard quotient ${\mathrm{THQ}}_{i,j}$ of five heavy metal pollutants in wheat.

#### 2.2.3. Total Carcinogenic Risk

Carcinogenic risk (CR) is commonly used to calculate the probability that an individual will develop a certain cancer as a result of exposure to a chemical [36,37,38,39], and the sum of the carcinogenic risks of various chemicals is the total carcinogenic risk (TCR) when multiple chemicals act on the body at the same time. In this paper, the total carcinogenic risk was used to assess the carcinogenic risk of heavy metals in wheat to humans through dietary intake, considering that wheat contains several heavy metal elements that can combine to cause carcinogenic effects in humans. Both the U.S. EPA and our Site Environmental Assessment Guidelines indicate that when TCR < 10^−6^, it means that the heavy metal contaminants in wheat have no carcinogenic risk to humans; when 10^−6^ < TCR < 10^−4^, it means that the risk of the contamination to human health is acceptable; when TCR > 10^−4^, it means that the human body is in a state of intolerable carcinogenic risk at the time. The formula for the daily dietary intake of individual heavy metals through wheat is shown below.
(5)$${\mathrm{EDI}}_{i,j}^{50}=\frac{F_{i}^{50}\times C_{\mathrm{avg}(i,j)}}{W}$$
where ${\mathrm{EDI}}_{i,j}^{50}$ denotes the daily intake of the jth heavy metal (mg/(kg·d)) through wheat in the diets of residents of province i. In addition, the meanings of $F_{i}^{50}$, $C_{\mathrm{avg}(i,j)}$ and W in the formula are as described in Section 2.2.2.
(6)$${\mathrm{TCR}}_{i}=\sum_{j}\frac{\mathrm{EF}\times \mathrm{ED}\times {\mathrm{CSF}}_{j}\times {\mathrm{EDI}}_{i,j}^{50}}{{\mathrm{AT}}_{C}}$$

${\mathrm{TCR}}_{i}$ denotes the total carcinogenic risk of four heavy metals consumed by residents of province i through wheat in their diets; ${\mathrm{AT}}_{C}$ denotes the duration of the carcinogenic effect (365 days/year×years of exposure, 70 years of exposure is assumed in this paper); and ${\mathrm{CSF}}_{j}$ (cancer slope factor) denotes the carcinogenic slope factor of heavy metal j (kg·d/mg). In addition, the meanings of EF and ED are as described in Section 2.2.2. EPA provides the carcinogenic slope factors for heavy metals: the CSFs for Cd, As, Pb and Cr are 6.3, 1.5, 0.0085 and 0.5 kg·d/mg, respectively.

### 2.3. Wheat Food Safety Risk Classification Based on K-Means++ Algorithm

Based on the above selected evaluation method for heavy metal contamination in wheat, a dataset of evaluation indicators for wheat samples, taken weekly from each province, is constructed, resulting in a three-dimensional space of evaluation indicators for wheat samples. In order to increase the daily supervision and risk monitoring of wheat’s food quality and safety, the wheat food safety risk level is graded, so as to denote a higher level of safety risk in the area and provide time to focus on supervision. To reduce the subjectivity and strong reliance on experience in food safety risk classification, this paper adopts a data-driven approach and uses clustering algorithms to automatically classify the three-dimensional space constructed above into reasonable risk classes. The data handled in this paper are of the numerical type and large volume, so the K-Means++ algorithm was selected, which can ensure better scalability and low algorithm complexity when dealing with large data sets while enabling more similarity between patterns in a cluster than between patterns not in the same cluster. The specific algorithmic steps are described below:Using the dataset of the wheat sample assessment indicator as input, a sample point in the dataset as randomly selected as the first initial clustering center.For each point x in the dataset, the distance D(x) between x and the cluster center was calculated.The point with the largest D(x) was selected as the new clustering center.Steps 2 and 3 were repeated until K clustering centers were selected.Using the K cluster centers calculated above as the initial cluster centers, the K-Means algorithm was run to cluster the dataset of wheat sample evaluation indices.

Finally, the three-dimensional space of the evaluation indicator constructed above was divided into K parts according to the K-Means++ algorithm, the points in the sample set were grouped into these K clusters and the clustering centers of these K clusters were calculated. According to the distance between the K cluster centers and the origin, these K clusters were divided into K food safety risk levels.

### 2.4. Pyraformer-Based Model for Predicting Food Safety Risk Levels of Wheat

Using the constructed dataset of wheat sample assessment indexes, a time series of each indicator in each province was constructed using the assessment indicators of historically sampled wheat samples. The model algorithm was used to predict each risk indicator of wheat in each province separately, based on historical data. Neural network models are widely used in medical detection, intelligent identification of agricultural pests and diseases and industrial equipment lifetime analysis, and food quality safety assessment is also characterized by time series prediction problems. Pyraformer neural network models [40] are widely used in various industries due to their powerful data reasoning and friendliness to long series processing; therefore, in this paper, we used a Pyraformer neural network for the prediction of food safety risk indicators of wheat. After completing the prediction of each indicator of wheat, the risk level of the wheat was predicted according to the aforementioned K-Means++ algorithm-based wheat food safety risk grading method.

#### 2.4.1. Wheat Food Safety Risk Indicators Forecast

In this paper, the Pyraformer neural network model was used to predict food safety risk indicators for wheat, and the model is shown in Figure 1. The Pyraformer neural network model improved upon the Transformer model by introducing global tokens to reduce the time complexity. In this paper, single-step prediction was used, as described in Section 2.4.2.

On the input side, we used the time series of each indicator $\left[{\mathrm{Indicator}}_{t},\ldots ,{\mathrm{Indicator}}_{t+i},\ldots {\mathrm{Indicator}}_{t+T}\right]$ and the number of weeks in a year $\left[{\mathrm{Week}}_{t},\ldots ,{\mathrm{Week}}_{t+i},\ldots {\mathrm{Week}}_{t+T}\right]$ as observations and covariates, respectively, and combined them with the location codes to sum them as inputs for the coarse scale construction module (CSCM).

The model uses CSCM to construct multi-resolution C meta-trees. The goal of CSCM is to initialize nodes at the coarser scales of the pyramid graph so that subsequent Pyramid Attention Modules (PAM) can exchange information between these nodes.

To further capture the temporal correlation of different ranges, PAM was introduced by passing messages using the attention mechanism in the pyramid diagram. Using pyramid diagrams to describe the temporal correlation of the observed time series in a multi-resolution manner, we were able to decompose the pyramid diagram into two parts: inter-scale connectivity and intra-scale connectivity. Inter-scale connections form a C-tuple, in which each parent node has C children. In addition, it is easier to capture long-range dependencies in coarser scales by simply connecting neighboring nodes via intra-scale connections.

Finally, single-step prediction was used in this study. After the sequence was encoded by PAM, the features given by the last node on all scales in the pyramid diagram were collected, and they were connected and inputted into the fully connected layer for the prediction output of each risk assessment indicator (${Indicator}_{t+T+1}$).

#### 2.4.2. Forecast of Food Safety Risk Levels for Wheat

In this paper, a Pyraformer-based wheat food safety risk level prediction model was constructed, as shown in Figure 2. The model consisted of 3 parts, which were the data processing layer, Pyraformer-based wheat sample risk indicator forecast layer and wheat food safety risk level prediction layer.

First, at the data processing level, weekly risk assessment indicators were constructed for each of the 20 provinces of the country over a 3-year period based on the aforementioned wheat food safety risk assessment method, and risk assessment indicators with time series characteristics were used to construct a food safety risk space for wheat, which was then combined with the aforementioned data-driven risk ranking method to rank the food safety risk space for wheat. In the data processing layer of Figure 2, the blue dashed line indicated the calculation of NIPI of wheat based on five heavy metals, the orange dashed line indicated the calculation of THQ of wheat based on five heavy metals, and the green dashed line indicated the calculation of TCR of wheat based on four heavy metals.

Secondly, in the Pyraformer-based wheat sample risk indicator forecast layer, each risk assessment indicator of each province was constructed as a time series of ${[\mathrm{NIPI}}_{t},\ldots ,{\mathrm{NIPI}}_{t+i},\ldots {\mathrm{NIPI}}_{t+T}]$, $\left[{\mathrm{THQ}}_{t},\ldots ,{\mathrm{THQ}}_{t+i},\ldots {\mathrm{THQ}}_{t+T}\right]$ and $\left[{\mathrm{TCR}}_{t},\ldots ,{\mathrm{TCR}}_{t+i},\ldots {\mathrm{TCR}}_{t+T}\right]$, and the time series was divided into a training set and a test set. The training set was used to train the parameters of the indicator prediction model, and the test set was used to test the accuracy of the indicator’s prediction. In this study, the data from week 1 to 138 of the 3 years were used as the training set, and the data from week 139 to 159 were used as the test set. T-t+1 was selected as the time window, and the time series of the three assessment indicators of each province were put into the Pyraformer neural network model. Then, the prediction model outputted the three food safety risk assessment indicators, ${\mathrm{NIPI}}_{t+T+1}$, ${\mathrm{THQ}}_{t+T+1}$ and ${\mathrm{TCR}}_{t+T+1},$ of each province at the moment of t+T+1.

Finally, in the wheat food safety risk level prediction layer, the three indicators predicted in the upper layer were compared with the cluster centers of the K food safety risk levels constructed above in terms of distance, and the cluster with the closest distance among them was selected to classify the current wheat sample risk level into the corresponding cluster, which indicated the predicted risk level.

## 3. Experimental Results and Discussion

### 3.1. Data Set of Evaluation Indicators for Wheat Samples

Based on the aforementioned safety risk evaluation method for wheat samples, this paper constructed an assessment indicator dataset, which was derived from wheat samples sampled in 20 provinces from 2019 to 2021. The three-dimensional attributes of each sample point in the dataset are the three safety risk assessment indicators of NIPI, THQ and TCR for a province in a certain week, respectively.

The 3-dimensional attribute values of the assessment indicator dataset are shown in Figure 3, Figure 4 and Figure 5, where the horizontal coordinates represent the number of weeks in 3 years and the vertical coordinates represent the value of each assessment index.

### 3.2. Ranking of Wheat Assessment Indicator Datasets

As can be seen in Figure 3, Figure 4 and Figure 5, the assessment indicators of the sampled wheat samples differed greatly in data magnitude in different dimensions, so the direct use of the original data for clustering was expected to result in the contribution of data of smaller magnitudes to clustering being ignored, which would have seriously affected the clustering effect. In this paper, the maximum–minimum method is used to normalize the dataset to eliminate the influence of different magnitudes on the clustering results and to make the data comparable. The formula is shown in Equation (7).
(7)$$x^{\prime }=\frac{x-\min}{\max-\min}$$
where $x^{\prime }$ denotes the normalized evaluation index, x denotes the evaluation indicator in the original data set, min denotes the minimum value in the dimension where x is located in the original data set and max denotes the maximum value in the dimension where x is located in the original data set. It can be seen from the formula that the normalized indicators all ranged from 0 to 1.

After normalizing the data, the data set was clustered using the K-Means++ algorithm. Since the clustering algorithm was an unsupervised machine learning algorithm, the optimal solution of the model could not be calculated, but only the local optimal solution of the model could be calculated. Therefore, in this paper, according to the actual demand of food safety risk classification, the K values of clustering centers were set from 2 to 6, respectively, and the number of clusters with the best clustering effect was selected by comparing the silhouette coefficients after clustering. Figure 6 shows the silhouette coefficients of each clustering result. The silhouette coefficient is a way to evaluate how good the clustering effect is, and aims to compare the similarity of an object to its own cluster with the similarity to other clusters. The number of clusters with the highest silhouette coefficient indicates the best choice of the number of clusters. As can be seen from Figure 6, when the number of clusters was selected as three, the silhouette coefficient was the largest, indicating that the clustering effect was the best at this time. Therefore, the risk level of the wheat assessment indicator dataset was classified as three in this paper.

After selecting the rank classification of the wheat assessment indicator dataset as three levels, the sample points in the space were clustered by the K-Means++ algorithm, and the normalized values of the sample centers of each cluster and the number of sample points in each cluster are shown in Table 1. According to the Euclidean distance between the normalized sample center and the origin, the three clusters were divided into three risk levels: high, medium and low. The specific distribution of points in the three clusters is shown in Figure 7, where the green points indicate low-risk wheat samples, the yellow points indicate medium-risk wheat samples and the red points indicate high-risk wheat samples. As seen from the figure, the higher the risk level was, the farther the sample points were from the origin in the three-dimensional space.

After dividing the points in the wheat sample set into risk classes, we conducted a probability density analysis of the original sample point distributions in each cluster, as shown in Figure 8, Figure 9 and Figure 10. From the figures, it can be seen that the concentration intervals, as well as the interval ranges, of the distribution of each dimensional attribute in each cluster differed significantly from each other, as follows:

(1)Figure 8 shows the probability density of each dimensional attribute of the low-risk cluster, where the values of NIPI are concentrated around 0.06 and mainly distributed from 0.03 to 0.09; the values of THQ are concentrated around 0.025 and mainly distributed from 0.01 to 0.15 and the values of TCR are concentrated around 0.05 × 10^−5^ and distributed from 0.025 × 10^−5^ to 0.3 × 10^−5^. Each indicator was distributed in a small range of values.(2)Figure 9 shows the probability density of each dimensional attribute of the medium-risk cluster, where the values of NIPI are concentrated around 0.2 and mainly distributed from 0.1 to 0.5; the values of THQ are concentrated around 0. 25 and mainly distributed from 0.15 to 0.4 and the values of TCR are concentrated around 0.75 × 10^−5^ and distributed from 0.05 × 10^−5^ to 1.75 × 10^−5^. The distribution of each indicator was still within a relatively small range of values, but its concentrated values and distribution areas were larger compared to the low-risk clusters.(3)Figure 10 shows the probability density of each dimensional attribute of the high-risk clusters, where the values of NIPI are concentrated around 1 and mainly distributed from 0.5 to 2.5; the values of THQ are concentrated around 1 and mainly distributed from 0. 5 to 1.5 and the values of TCR are concentrated around 3 × 10^−5^ and distributed from 2 × 10^−5^ to 4 × 10^−5^. The distribution of each indicator was in a relatively large range of values, and the concentrations of values and distribution areas were larger compared with the other two clusters.

### 3.3. Risk Level Prediction for Wheat Samples

Based on the prediction model of wheat food safety risk level constructed above, this experiment firstly predicted three risk assessment indicators for wheat in 20 provinces. In Figure 11, Figure 12, Figure 13, Figure 14 and Figure 15, the dashed line at week 138 indicates the dividing line, the data from week 138 and earlier indicate the training set and the data from week 139 and later indicate the test set. In the figure, the solid blue line represents the actual data trend of the constructed evaluation metrics for the training of the model, the solid yellow line before the dashed line indicates the data trend trained by the model and the dashed yellow line after the dashed line indicates the predicted data trend.

According to the range of each indicator, described in Section 2.2, that is allowed when there is no contamination or less contamination, combined with the data trend in Figure 11, Figure 12, Figure 13, Figure 14 and Figure 15, it can be seen that, overall, China’s wheat was less contaminated by heavy metals and less harmful to human health; however, some individual provinces in some time range indicators exceeded the standard. However, it is still necessary to draw the attention of relevant government departments and to take timely regulatory measures to avoid serious food safety risk problems.

As seen in Figure 11, the THQ of wheat in Beijing exceeded the standard in June at the time of 2020, and the NIPI and TCR did not exceed the standard, indicating that the chronic risk of heavy metal contamination in wheat was higher in that month. Considering that at the time, new wheat was available and flowing into Beijing from all over the world, it was necessary to increase the supervision and sampling of wheat in Beijing in June, and, at the same time, to conduct food traceability of foreign wheat flowing into Beijing so as to strengthen the supervision of wheat in the field.

In Figure 12, in early February of each year during the period of 2019–2021, NIPI and THQ indicators in Henan Province exceeded the standard, while TCR did not; meanwhile, NIPI indicators in Hubei Province exceeded the standard, while THQ and TCR did not. Through the research, it was found that during this time period, due to the high demand for holiday replenishment, the escalation of epidemic prevention and control, the increase in the placement of grain reserves at all levels and multiple other factors, in two provinces, wheat sampling indicated that heavy metal contamination was more serious, and this period required strict control of wheat food safety. At the same time, it was found that although the indicators of heavy metal pollution in wheat in Heilongjiang Province were within the permissible range, the deviation of the predicted value from the true value was significant, taking into account that the province may have been increasing industrial production and other projects, thus causing heavy metal pollution in the soil.

In Figure 13 and Figure 14, all of the assessed indicators of heavy metal contamination of wheat in Hunan, Jilin, Jiangsu, Jiangxi, Liaoning, Inner Mongolia, Ningxia and Qinghai provinces were at low levels, indicating that these eight provinces were doing a relatively good job ensuring the food safety of wheat. However, it was found that the risk indicators in Jilin province, although within the allowed range, also showed large fluctuations, requiring further analysis of heavy metal contamination of the soil, air and other environments.

As seen in Figure 15, there was an excess of heavy metals in wheat in Shaanxi and Sichuan. The NIPI and THQ indicators in Shaanxi province exceeded the standard at the beginning of February each year, while the TCR indicator fluctuated more significantly at that time, as did the NIPI in Sichuan province. As both Shaanxi and Sichuan are large pasta provinces, they were in the same situation as Henan; therefore, it was also necessary to increase the supervision of wheat in the province during this period.

The predictions in graphs 11 to 15 show that the actual trends of most of the data were basically the same as the predicted trends, and only individual predicted data showed large deviations from the actual data. Therefore, the effectiveness of the prediction model in this experiment was evaluated by calculating the RMSE and MAE of the predicted indicators for each province based on the predicted data and the actual data. Figure 16 and Figure 17 show the root mean square error (RMSE) and mean absolute error (MAE) of the model, predicting the 3 indicators for 20 provinces across the country, respectively. Since the allowable range of TCR indicators is very small, the order of magnitude is in the range of 1 × 10^−4^–1 × 10^−6^. In order to better demonstrate the prediction effect of the model on TCR indicators, the RMSE and MAE indicators of TCR were multiplied by E5, and the prediction model efficacy indicators of NIPI and THQ were kept in the same order of magnitude. The smaller the RMSE and MAE, the smaller the prediction error and the better the effect. As can be seen in Figure 16 and Figure 17, the RMSE and MAE of each risk assessment indicator predicted by the model for each province were kept below 1. Thus, this experiment achieved better results in the process of indicator prediction.

In addition, to verify the effectiveness of this model in predicting the risk level of heavy metal contamination in wheat, risk level prediction models based on the long short-term memory (LSTM), gate recurrent unit (GRU) and Informer neural networks are used in this paper for comparison with this model, and their precision and recall rates of the F1 values of each model are shown in Table 2. From the table, it can be seen that the risk level prediction model based on the Pyraformer neural network proposed in this paper achieved the best results in terms of precision, recall and F1 values for different levels of risk prediction. Precision, recall and F1 values are shown in Table 2 as P%,R%,F1% respectively. Due to the different numbers of wheat samples of low, medium and high levels in the sample, especially the fact that the lowest number of samples were found at high levels, the precision and recall of this model still reached more than 90%, far exceeding the other models by 25.38–4.15% and 18.42–5.26%, respectively. This shows that the risk level prediction model proposed in this paper is able to meet the current early warning function for the risk of heavy metals in wheat.

## 4. Conclusions

Among China’s food crop varieties, wheat is the main food crop, second only to rice, and is cultivated in a wide range of areas in China, from the tropical Hainan Island in the south to the cold region of Heilongjiang in the north. In recent years, wheat production has increased significantly due to farmers’ choices of good varieties and timely work related to wheat pests and disease prevention. The relevant departments provide year-round coverage regulation of the nation’s wheat, which can lead to the occupation of sampling forces and waste of management resources. This results in a lack of focused supervision of high-risk areas, causing frequent food safety problems, triggering serious declines in public opinion and bringing about adverse social impacts. Therefore, predicting the level of heavy metal contamination of wheat in China, grading the risk of wheat food safety and focusing supervision on areas with serious contamination levels can effectively prevent food safety problems and enhance people’s sense of well-being and security, while providing a scientific basis for dynamic monitoring and comprehensive prevention of heavy metal contamination of wheat in agricultural fields. From the above experimental process, we can also see that the predicted future data are inferred from the historical and current data; therefore, the comprehensiveness and accuracy of the sampling data directly affected the prediction results.

China is a large agricultural country. Its five major varieties of grain crops are rice, wheat, corn, soybeans and potatoes, in addition to sorghum, cereals and other mixed crops. In China, grain is both a necessity for people’s survival and an important source of feed, as well as an important economic source for food and agriculture farmers. Therefore, the food safety of grain is not only important for stabilizing people’s psychology, but also significant to the long-term stable development of the national economy. By applying the food safety risk prediction model proposed in this paper to various food categories, risk assessment and prediction of heavy metals in food can improve our ability to resist food risks and reduce the social crisis derived from food risks.

## Figures and Tables

**Figure 1 foods-12-01843-f001:**
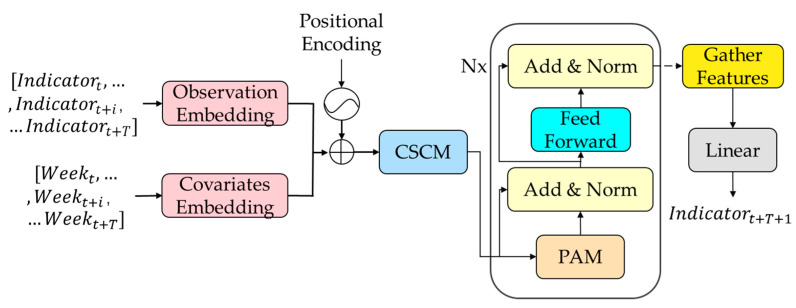
Pyraformer-based model for predicting food safety risk indicators for wheat.

**Figure 2 foods-12-01843-f002:**
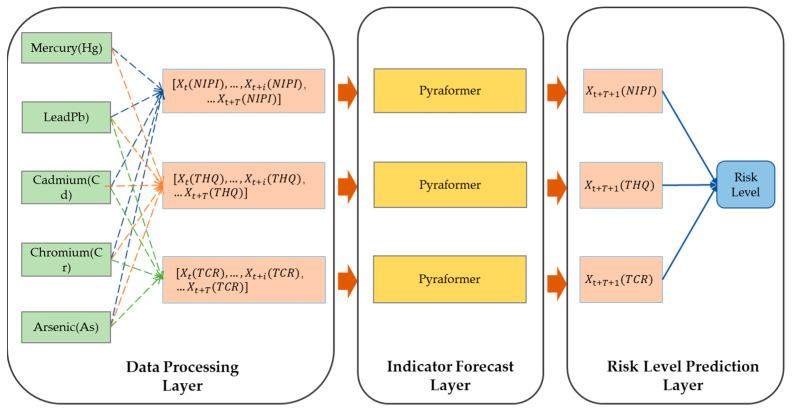
Pyraformer-based model for predicting food safety risk levels in wheat.

**Figure 3 foods-12-01843-f003:**
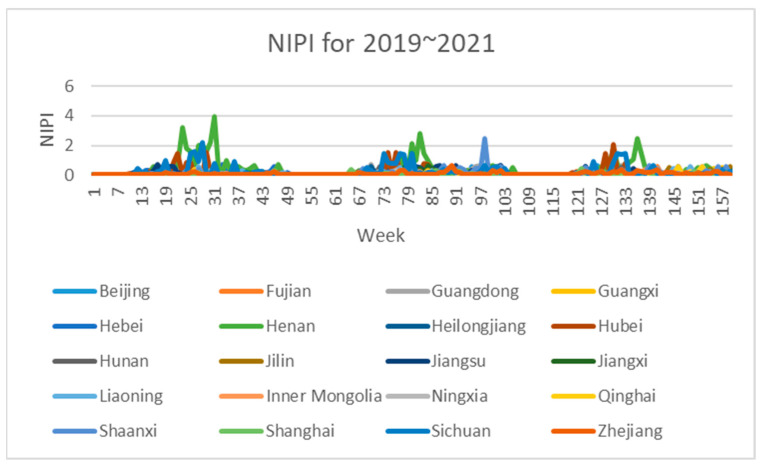
Dataset of wheat sample assessment indicator NIPI.

**Figure 4 foods-12-01843-f004:**
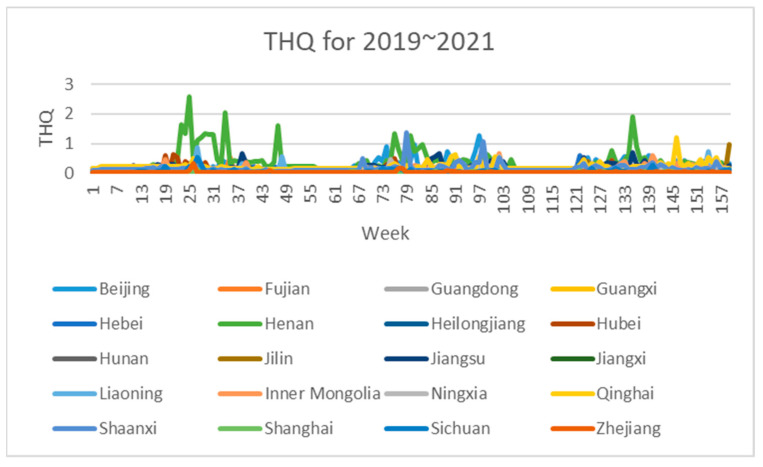
Dataset of wheat sample assessment indicator THQ.

**Figure 5 foods-12-01843-f005:**
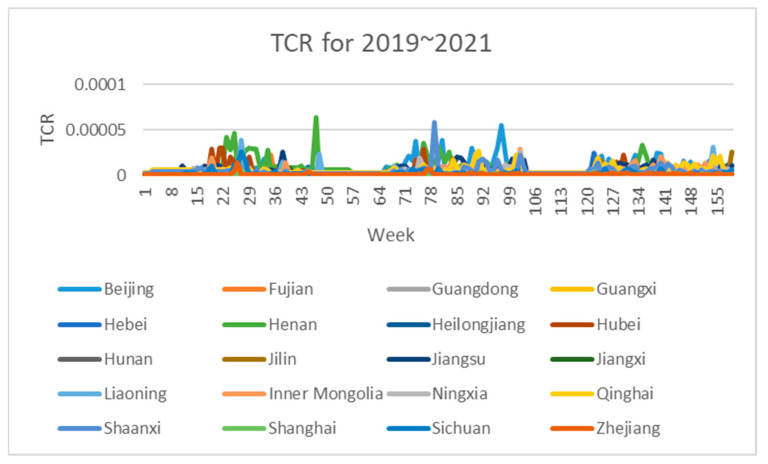
Dataset of wheat sample assessment indicator TCR.

**Figure 6 foods-12-01843-f006:**
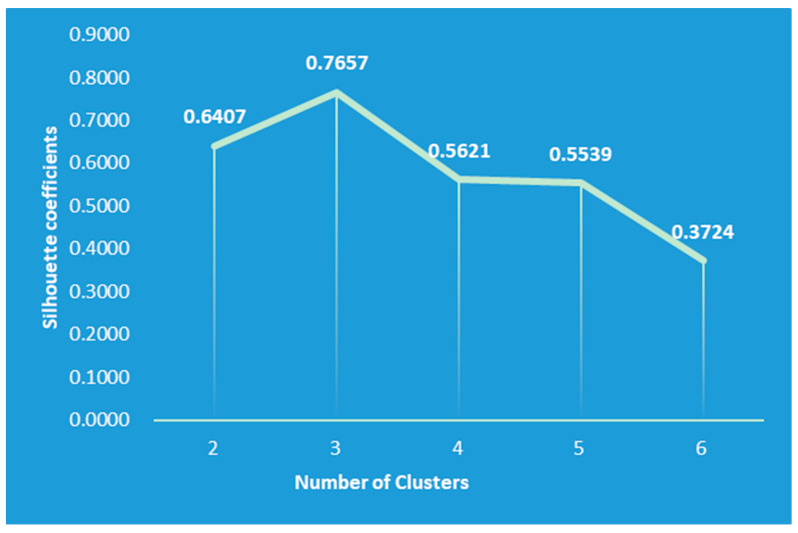
Silhouette coefficient of each clustering result.

**Figure 7 foods-12-01843-f007:**
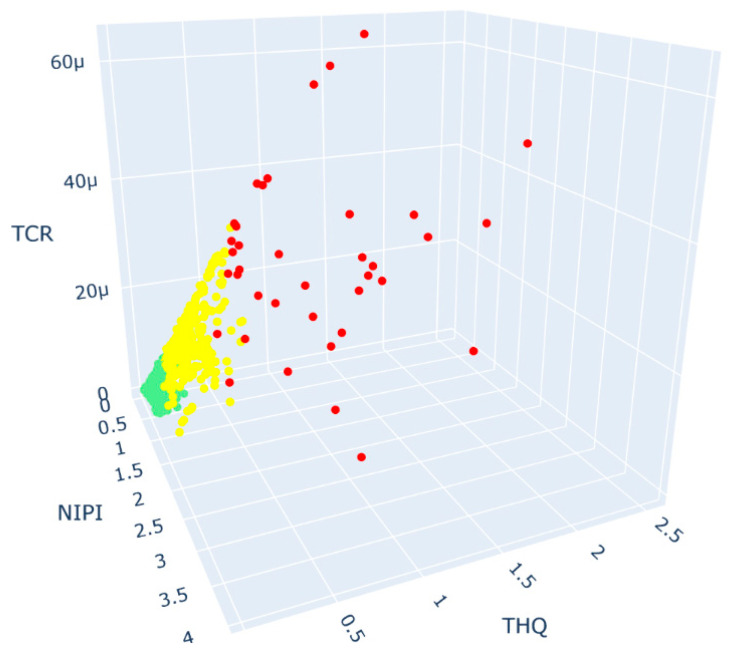
Three-dimensional space plot of K-Means++ clustering results.

**Figure 8 foods-12-01843-f008:**
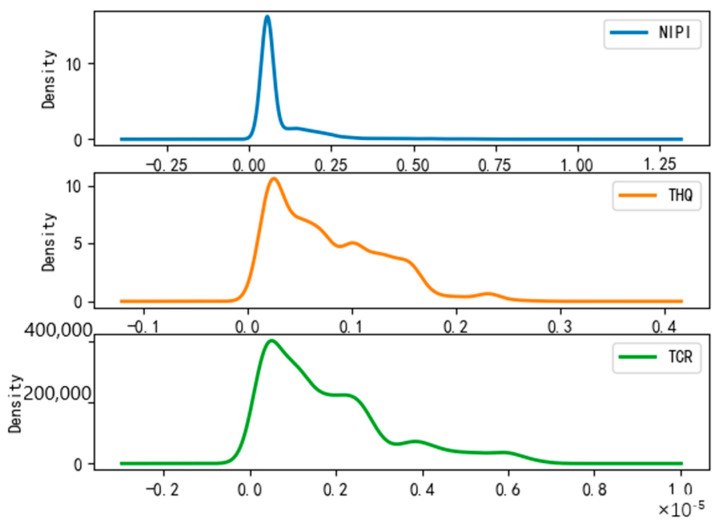
Probability density plot of low-risk clusters.

**Figure 9 foods-12-01843-f009:**
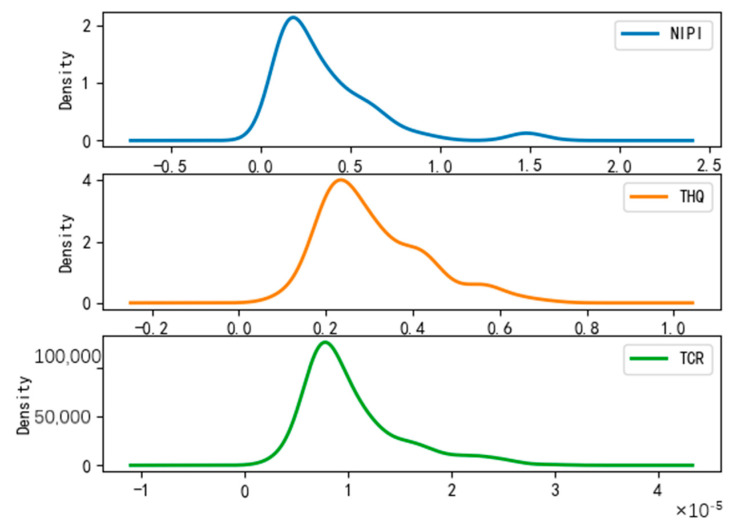
Probability density plot of medium-risk clusters.

**Figure 10 foods-12-01843-f010:**
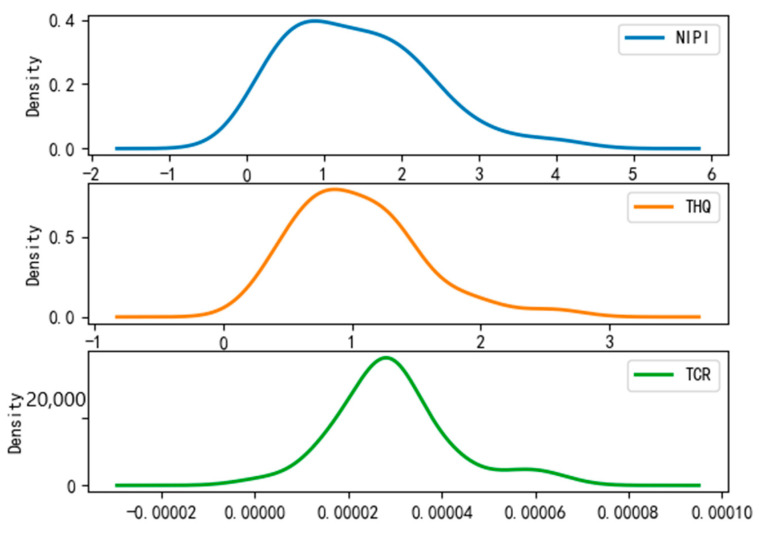
Probability density plot of high-risk clusters.

**Figure 11 foods-12-01843-f011:**
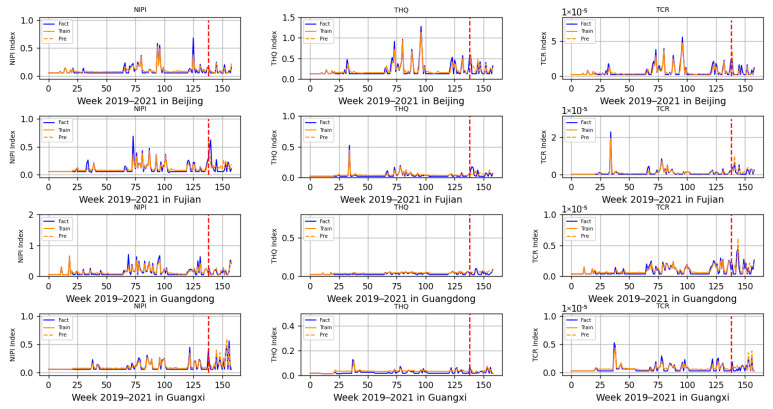
Risk assessment indicator predictions for Beijing, Fujian, Guangdong and Guangxi.

**Figure 12 foods-12-01843-f012:**
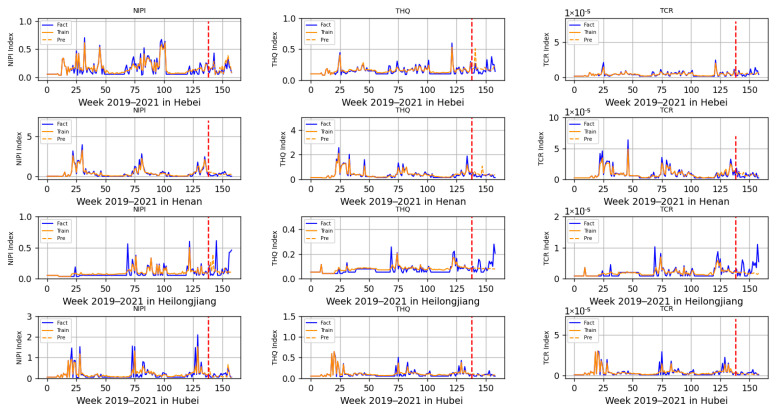
Risk assessment indicator predictions for Hebei, Henan, Heilongjiang and Hubei.

**Figure 13 foods-12-01843-f013:**
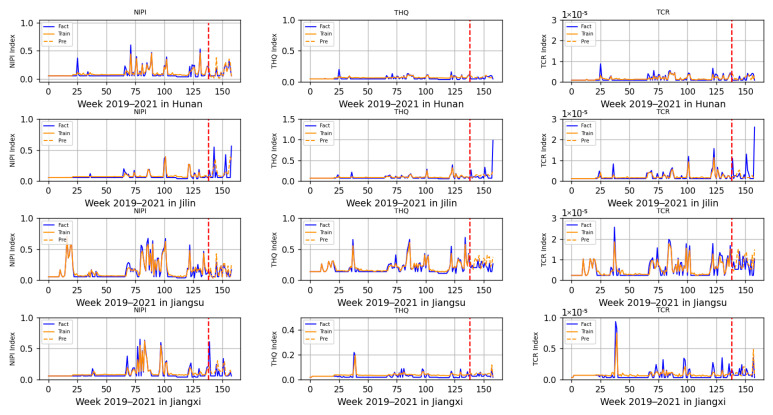
Risk assessment indicator predictions for Hunan, Jilin, Jiangsu and Jiangxi.

**Figure 14 foods-12-01843-f014:**
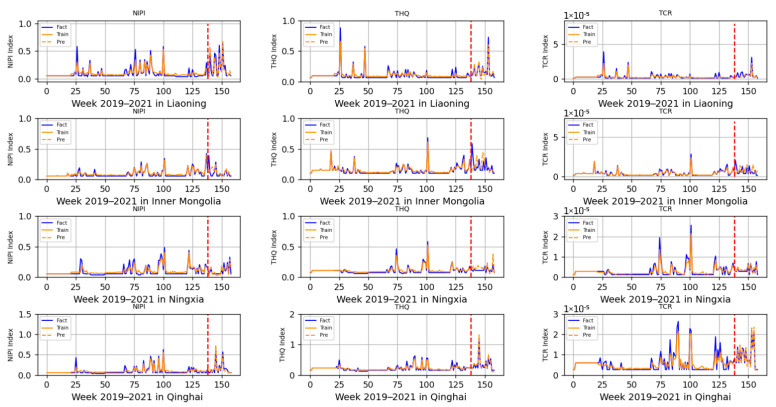
Risk assessment indicator predictions for Liaoning, Inner Mongolia, Ningxia and Qinghai.

**Figure 15 foods-12-01843-f015:**
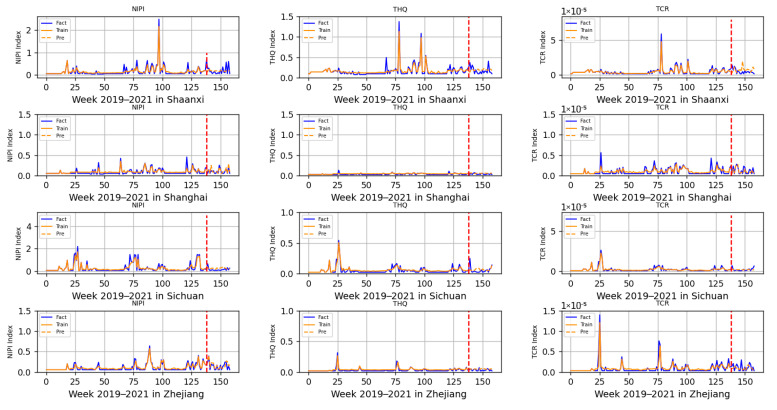
Risk assessment indicator predictions for Shaanxi, Shanghai, Sichuan and Zhejiang.

**Figure 16 foods-12-01843-f016:**
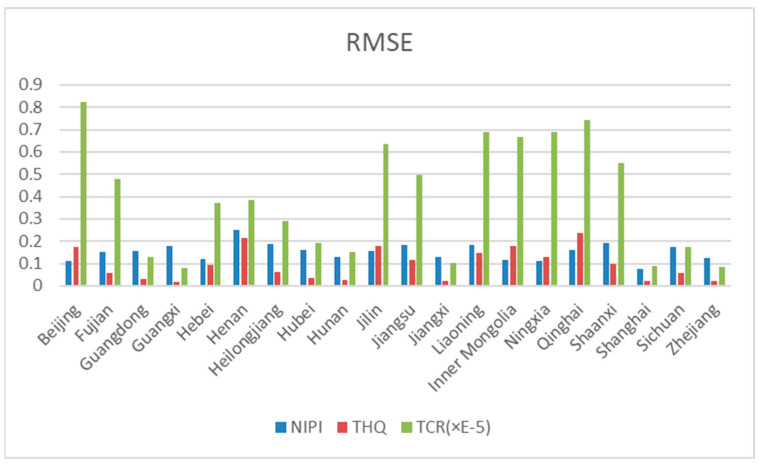
RMSE as a metric for assessing model effectiveness.

**Figure 17 foods-12-01843-f017:**
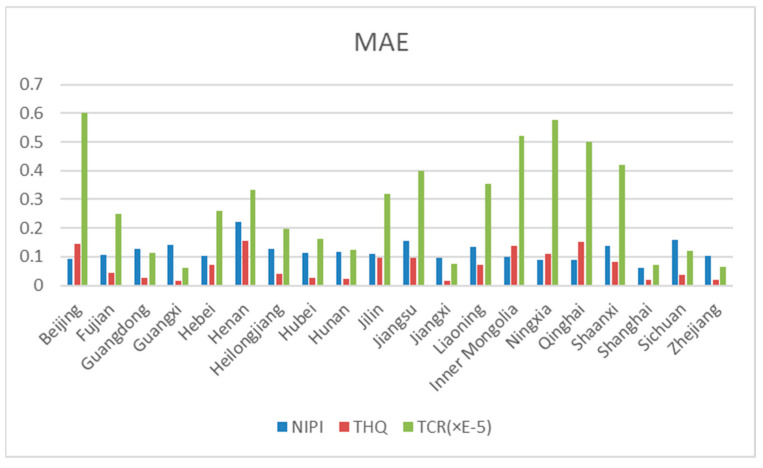
MAE as a metric for assessing model effectiveness.

**Table 1 foods-12-01843-t001:** Clustering centers, sample points and rank levels.

Cluster	NIPI	THQ	TCR	Sample Point	Rank Level
1	0.013764	0.02515	0.02412	2771	Low
2	0.079831	0.115745	0.157608	371	Medium
3	0.35066	0.403727	0.461727	38	High

**Table 2 foods-12-01843-t002:** Evaluation metrics for neural network-based risk level prediction models.

Model	Low Level			Medium Level			High Level		
	P%	R%	F1%	P%	R%	F1%	P%	R%	F1%
LSTM	98.83	97.94	98.39	85.35	89.49	87.37	66.67	78.95	72.29
GRU	98.95	98.16	98.55	86.86	90.84	88.80	76.74	86.84	81.48
Informer	99.13	98.52	98.82	89.56	92.45	90.98	83.33	92.11	87.50
Pyraformer	99.24	98.81	99.02	91.41	94.61	92.98	92.50	97.37	94.87

## Data Availability

Restrictions apply to the availability of these data. Data were obtained from the State Administration for Market Regulation Statistics and are available at [41] with the permission of the State Administration for Market Regulation Statistics.

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
