# Peer review of "Prediction of Food Safety Risk Level of Wheat in China Based on Pyraformer Neural Network Model for Heavy Metal Contamination"

_foods, 2023, doi:10.3390/foods12091843_

Round 1

Reviewer 1 Report

Thank you for the opportunity to review this article.

The authors evaluated the Prediction of Food Safety Risk Level of Wheat in China Based on Pyraformer Neural Network Model for Heavy Metal Contamination.

Some minor remarks follow.

Line 73.6.32x10(-4)....???

Line 134-135. Please use the dame font

Line 143.jth heavy metal contaminant....what is jth...please explain more.

Line 149. Pavg2...not explained in the text.

Line 152.Pmin.. do not exist in the equation 2

Please write all the references with the same way and according journal's guidelines.

The Quality of English Language is fine.

Reviewer 2 Report

Dear authors,

I've reviewed your manuscript and I'm impressed with the clarity of the objectives and rationale of your study, which aims to develop a Pyraformer-based model to predict the safety risk level of Chinese wheat contaminated with heavy metals. The study is well-motivated by the need for hierarchical management and early warning of heavy metal contamination in wheat, a significant food crop in China.

Your paper presents a comprehensive and well-structured methodology, making it a valuable contribution to the field. While the methods section provides sufficient information for replication, it's important to acknowledge that reproducing a study of this scale may be challenging due to the significant resources required. In light of these challenges, your paper reads more like a case study, offering insights and lessons for researchers interested in conducting similar investigations. Kudos for your rigorous methodology and for sharing your experiences, which can help inform and guide future research in this area.

Regarding the conclusions section, I suggest addressing the following points to improve clarity and robustness:

  1. Future research: Expand the conclusions to include recommendations for future research or monitoring efforts in the area of heavy metal contamination in wheat and other food crops. This will help guide future studies and policy interventions.
  2. International comparison: Compare China's heavy metal contamination levels in wheat with those of other countries to provide a global context. This comparison could help identify best practices and lessons learned from other regions applicable to China.
  3. Limitations and biases: Address any potential biases or limitations in the data collection and analysis process that could impact your study's conclusions. Discussing these limitations will strengthen your manuscript and ensure the validity of your results.
  4. Policy implications and recommendations: Provide specific policy recommendations or interventions that could help reduce heavy metal contamination in wheat production based on your study's findings.

Your paper is overall very good and provides valuable insights into the topic of heavy metal contamination in wheat production. Addressing these points in the conclusions section will make your manuscript even stronger and more comprehensive. I look forward to seeing the revised version of your manuscript.

The English language usage in your manuscript is generally good, but there is room for improvement in grammar, syntax, consistency in terminology, and clarity. I recommend having a native English speaker or a professional language editor review your manuscript to address these issues.

By making these language-related improvements, your manuscript will be more accessible to a broader audience and better suited for publication.
